# Mitochondrial fusion but not fission regulates larval growth and synaptic development through steroid hormone production

Hector Sandoval[1], Chi-Kuang Yao[1†], Kuchuan Chen[2], Manish Jaiswal[1,3], Taraka Donti[1], Yong Qi Lin[3], Vafa Bayat[2,4‡], Bo Xiong[2§], Ke Zhang[5¶], Gabriela David[2], Wu-Lin Charng[1,2], Shinya Yamamoto[1,2,6], Lita Duraine[3], Brett H Graham[1,2], Hugo J Bellen[1,2,3,6,7]*

[1]Department of Molecular and Human Genetics, Baylor College of Medicine, Houston, United States; [2]Program in Developmental Biology, Baylor College of Medicine, Houston, United States; [3]Howard Hughes Medical Institute, Baylor College of Medicine, Houston, United States; [4]Medical Scientist Training Program, Baylor College of Medicine, Houston, United States; [5]Program in Structural and Computational Biology and Molecular Biophysics, Baylor College of Medicine, Houston, United States; [6]Jan and Dan Duncan Neurological Research Institute, Texas Children's Hospital, Houston, United States; [7]Department of Neuroscience, Baylor College of Medicine, Houston, United States

*For correspondence: hbellen@bcm.edu

Present address: †Institute of Biological Chemistry, Academia Sinica, Taipei, Taiwan; ‡Department of Pathology, Stanford Hospital and Clinics, Stanford, United States; §Department of Genome Sciences, University of Washington, Seattle, United States; ¶Department of Neurology, Johns Hopkins University School of Medicine, Baltimore, United States

Competing interests: The authors declare that no competing interests exist.

**Abstract** Mitochondrial fusion and fission affect the distribution and quality control of mitochondria. We show that Marf (Mitochondrial associated regulatory factor), is required for mitochondrial fusion and transport in long axons. Moreover, loss of *Marf* leads to a severe depletion of mitochondria in neuromuscular junctions (NMJs). *Marf* mutants also fail to maintain proper synaptic transmission at NMJs upon repetitive stimulation, similar to *Drp1* fission mutants. However, unlike *Drp1*, loss of *Marf* leads to NMJ morphology defects and extended larval lifespan. Marf is required to form contacts between the endoplasmic reticulum and/or lipid droplets (LDs) and for proper storage of cholesterol and ecdysone synthesis in ring glands. Interestingly, human Mitofusin-2 rescues the loss of LD but both Mitofusin-1 and Mitofusin-2 are required for steroid-hormone synthesis. Our data show that Marf and Mitofusins share an evolutionarily conserved role in mitochondrial transport, cholesterol ester storage and steroid-hormone synthesis.

## Introduction

Mitochondrial dynamics plays a critical role in the control of organelle shape, size, number, function and quality control of mitochondria from yeast to mammals (*Westermann, 2009*; *Chan, 2012*). It consists of fusion and fission of mitochondria, which are regulated by several GTPases (*van der Bliek et al., 2013*). Mitochondrial fusion requires the fusion of the outer membrane followed by inner membrane fusion (*Chan, 2012*; *Mishra et al., 2014*). In mammals, Mitofusin 1 (Mfn1) and Mitofusin 2 (Mfn2) regulate outer mitochondrial fusion whereas inner membrane fusion is controlled by Optic atrophy protein 1 (Opa1). Mitochondrial fission is regulated by Dynamin related protein 1 (Drp1) (*van der Bliek et al., 2013*). Decreased fusion results in fragmented round mitochondria, while defective fission leads to fused and enlarged mitochondria (*van der Bliek et al., 2013*).

**eLife digest** Mitochondria are the main source of energy for cells. These vital and highly dynamic organelles continually change shape by fusing with each other and splitting apart to create new mitochondria, repairing and replacing those damaged by cell stress.

For nerve impulses to be transmitted across the gaps (called synapses) between nerve cells, mitochondria need to supply the very ends of the nerve fibers with energy. To do this, the mitochondria must be transported from the main body of the nerve cell to the tips of the nerve fibers. This may not happen if mitochondria are the wrong shape, size or damaged.

While searching for genetic mutations that disrupt nerve function in the fruit fly *Drosophila*, Sandoval et al. spotted mutations in a gene called *Marf*. Further investigations revealed that flies with mutant versions of *Marf* have small, round mitochondria, and their nerves cannot transmit signals to muscles when they are highly stimulated. This is because the mutant mitochondria are not easily transported along nerve fibers, and so not enough energy is supplied to the synapses. The synapses of the *Marf* mutants are also abnormally shaped. Sandoval et al. found that this is not because *Marf* is lost in the neurons themselves, but because it is lost from a hormone-producing tissue called the ring gland.

Another problem found in flies with mutated *Marf* genes is that they stop developing while in their larval stage. Sandoval et al. established that this could also be related to the loss of *Marf* from the ring gland. The Marf protein has two different functions in the ring gland: forming and storing droplets of fatty molecules used in hormone production, and synthesising a hormone that controls when a fly larva matures into the adult fly. This suggests that the lower levels of this hormone produced by *Marf* mutant flies underlies their prolonged larval stages and synapse defects.

Vertebrates (animals with backbones, such as humans) have two genes that are related to the fly's *Marf* gene. When the human forms of these genes were introduced into mutant flies that lack a working copy of *Marf*, hormone production was only restored if both genes were introduced together. This indicates that these genes have separate roles in vertebrates, but that these roles are both performed by the single fly gene.

The role of *Marf* in tethering mitochondria in the ring gland may allow us to better understand how this process affects hormone production and how the different parts of the cell communicate.

Loss of these mitochondrial GTPases results in lethality in worms, flies and mice (*Chen et al., 2003*; *Westermann, 2009*; *Debattisti and Scorrano, 2012*). Mutations in the human *DRP1* gene causes a dominant fatal infantile encephalopathy associated with defective mitochondrial and peroxisomal fission (*Waterham et al., 2007*). On the other hand, missense mutations in *OPA1* lead to a dominant optic atrophy (*Alexander et al., 2000*; *Delettre et al., 2000*). Depending on the severity of the mutation, patients may also suffer from ataxia and neuropathy (*Yu-Wai-Man et al., 2010*). Also, missense mutations in *MFN2* cause Charcot-Marie-Tooth type 2A, a common autosomal dominant peripheral neuropathy associated with axon degeneration (*Zuchner et al., 2004*). Finally, aberrant levels of mitochondrial GTPases have been associated with Parkinson's, Huntington's and Alzheimers' diseases (*Itoh et al., 2012*). These observations in model organisms and human patients suggest that mitochondrial dynamics affects neuronal maintenance in many different contexts.

A significant imbalance of mitochondrial fission and fusion may affect the subcellular distribution of mitochondria, especially in neurons since they need to efficiently traffic from the soma to the synapses (*Sheng, 2014*). Loss of *Drosophila Drp1* impairs the delivery of mitochondria to neuromuscular junctions (NMJs), likely because they are large and interconnected. This defect is also associated with a severe depletion of mitochondria in NMJs, which affects local ATP production. This in turn affects the trafficking of synaptic vesicles upon endocytosis during prolonged stimulation (*Verstreken et al., 2005*). Similarly, in vertebrates, loss of *Drp1* leads to an accumulation of mitochondria in the soma and reduced mitochondrial density in dendrites of hippocampal neurons (*Li et al., 2004*). The *Drp1* data in flies and vertebrates indicate that the expanded size of mitochondria affects their mobility (*Sheng, 2014*).

Mitochondrial trafficking may also be affected by the physical interaction between the mitochondria and the transport machinery. Recent studies have documented a direct interaction between Mfn2 and a motor adaptor complex for mitochondrial transport, Miro2 (*Misko et al., 2010*). Moreover, loss of *MFN2* in Purkinje cells displayed reduced mitochondrial motility in cerebellar dendrites (*Chen et al.,*

*2007*) and reduced mitochondrial transport in axons in cultured dorsal root ganglion neurons (*Misko et al., 2010*). These data suggest that an interaction of Mfn2 with Miro2 may be important for its role in trafficking (*Misko et al., 2010*). Although loss of both *Drp1* and *MFN2* impair mitochondrial trafficking, a careful comparison of the phenotypes associated with loss of *Drosophila Drp1*, Mitofusin or *Marf*, would be useful as the suggested mechanisms by which they impair transport seem very different.

In addition to their roles in fission and fusion, Drp1, Mfns and Opa1 have been implicated in a variety of other processes. For example, Drp1 has been shown to facilitate the induction of apoptosis (*Frank et al., 2001*) whereas Opa1 was shown to affect the stability of cristae junction in inner mitochondrial membrane (*Frezza et al., 2006*). Finally, Mfn2 also tethers mitochondria to the endoplasmic reticulum (ER) to mediate $Ca^{2+}$ uptake (*de Brito and Scorrano, 2008*). However, the molecular mechanisms underlying these non-canonical functions are less well studied.

In an unbiased screen designed to identify essential genes that affect neuronal function (*Yamamoto et al., 2014*), we identified the first mutant allelic series of *Marf* in *Drosophila*. Here we exploit these mutants to determine how loss of *Marf* affects mitochondrial transport when compared to *Drp1* loss. Surprisingly, we observe NMJ defects only in *Marf* mutants but not in *Drp1* mutants. These defects are regulated non-cell autonomously by steroid-hormones produced in ring glands (RG), a major endocrine organ in insects. Through expression of human *MFN1* or *MFN2* in *Marf* mutant RG, we show that MFN1 and MFN2 have both distinct and complementary roles.

## Results

### *Marf* affects mitochondrial distribution in photoreceptors

Through a forward genetic screen on the *Drosophila X*-chromosome (*Yamamoto et al., 2014*) we isolated seven independent lethal alleles of *Marf* that affect electroretinogram (ERG) recordings in homozygous mutant clones (*Figure 1A,C*, *Figure 1—figure supplement 1*). The on- and off-transients (*Figure 1A*, red arrows) of the ERG are a read-out of synaptic transmission between photoreceptors (PR) and postsynaptic cells, while the amplitude of the depolarization (*Figure 1A*, green bracket) is a measure of the function of the phototransduction cascade (*Wang and Montell, 2007*). The *Marf* mutations vary in strength (*Figure 1A,E* and *Figure 1—figure supplement 1B*), providing an allelic series. ERG recordings in homozygous mutant eye clones reveal a reduction in on- and off-transients as well as loss of amplitude in one day old flies (*Figure 1A*). The ERG recordings differ from *Drp1* mutants that only exhibit a loss of on- and off-transients but a normal amplitude (*Figure 1B*, [*Verstreken et al., 2005*]). In summary, loss of *Marf* severely impairs the phototransduction cascade as well as synaptic transmission, whereas loss of *Drp1* mainly affects synaptic transmission of PRs.

Lethal staging shows that most *Marf* mutants (*A, B, E, F* and *G*) die as third instars after a very extended larval stage period of 18–21 days, which typically takes 6 days in wild type animals (*Figure 1—figure supplement 1B*). The lethality of all *Marf* mutants is rescued by a *Marf* genomic DNA construct or by a ubiquitously expressed *Marf* cDNA (*Figure 1—figure supplement 1A*), showing that the *Marf* mutations are responsible for the lethality (*Figure 1—figure supplement 1B*). Moreover, transheterozygous *Marf^B^/Df(1)Exel6239* female mutants display the same lethal phase as *Marf^B^/Y* males, suggesting that *Marf^B^* is likely to be a severe loss of function allele or null allele (*Figure 1—figure supplement 1B*). Finally, *Marf^B^* hemizygous males exhibit a severe protein loss compared to *Marf^G^* hemizygous males and controls (*Figure 1—figure supplement 1C*), suggesting that this missense mutation in the GTPase domain (*Figure 1C*) also destabilizes the protein.

Since mitochondrial transport has been shown to be affected in some neurites of *MFN2*-deficient vertebrate cells (*Chen et al., 2007*), we performed Transmission Electron Microscopy (TEM) at the PR terminals. *Marf* mutants exhibit a very severe loss of mitochondria (*Figure 1D*, yellow arrows) in PR terminals when compared to control (*Figure 1D,E*). The severity of the loss of mitochondria (*Figure 1E*) correlates with the loss of neuronal function gauged by ERGs (*Figure 1A*). These data are reminiscent of the documented lack of mitochondria in PR terminals in *Drp1* mutants (*Verstreken et al., 2005*). However, the mitochondria in *Marf* mutant PRs are significantly smaller in size than controls (*Figure 1D*, yellow arrows), suggesting that an active transport mechanism is impaired.

### *Marf* affects mitochondrial function and distribution in NMJs

To assess if mitochondrial size is also affected in mutant muscles, we stained *Marf* and *Drp1* (*Figure 2— source data 1*) mutants with an anti-mitochondrial complex V antibody (ATP5A) (*Baqri et al., 2009*).

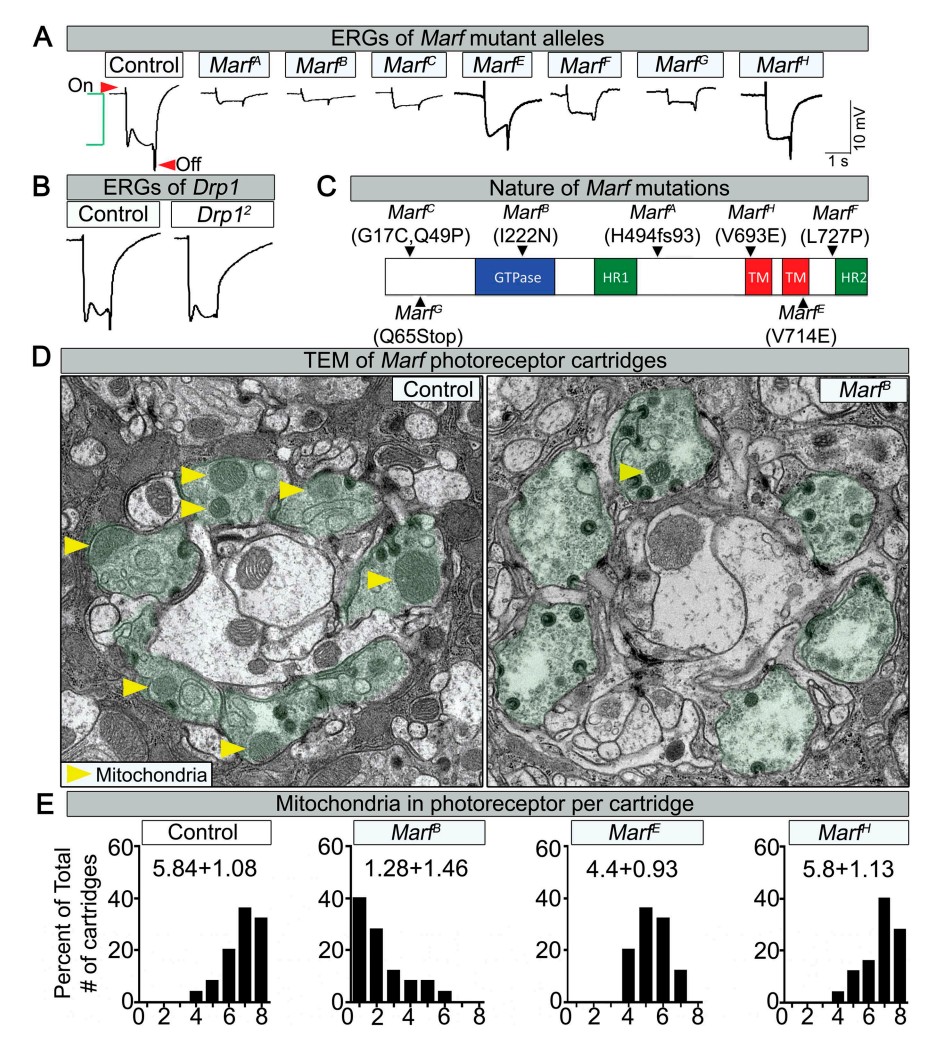

**Figure 1**. Loss of *Marf* impairs phototransduction and affects mitochondrial localization to photoreceptor terminals. (**A**) Electroretinograms (ERGs) of 1 day old ey-FLP mutant clones of 7 different *Marf* mutants or isogenized wild type clones (Control). ERGs of *Marf* mutant alleles and control flies. A typical ERG trace is comprised of an on-transient (red arrow), a depolarization (green bracket) and an off-transient (red arrow). (**B**) ERGs of *Drp1* mutants and control flies. (**C**) Marf protein domains and localization of EMS-induced mutations of the seven *Marf* mutant alleles identified by sequencing. H494fs93 = insertion of an **A** at nucleotide codon for amino acid H494 that generates 93 new amino acids followed by a premature stop codon. TM = transmembrane domain. HR = heptad repeat. (**D**) TEM sections of a cartridge containing fly photoreceptor terminals (green shading). *Marf* mutant photoreceptor terminals display reduced number and size of mitochondria (yellow arrow heads) compared to *Marf*-genomic rescue controls. (**E**) Quantification of total mitochondria number per cartridge in *Marf* mutants and *Marf*-genomic rescue photoreceptor terminals (Control). 50 cartridges per genotype.

The following figure supplement is available for figure 1:

**Figure supplement 1**. Mapping, lethal staging and Marf protein expression of Marf mutant alleles.

As expected, *Drp1* mutants have filamentous mitochondria whereas *Marf* mutants have small, rounded mitochondria (***Figure 2A*** and ***Figure 2—source data 2***). However, both *Marf* and *Drp1* mutant mitochondria produce similar reduced levels of ATP when compared to controls (***Figure 2C*** and ***Figure 2—source data 2***). Interestingly, the mitochondrial membrane potential (MMP) of *Drp1* mutants as measured with tetramethylrhodamine ethyl ester (TMRE) (***Scaduto and Grotyohann, 1999***) is slightly elevated, as reported before (***Verstreken et al., 2005***), when compared to controls whereas MMP of *Marf*

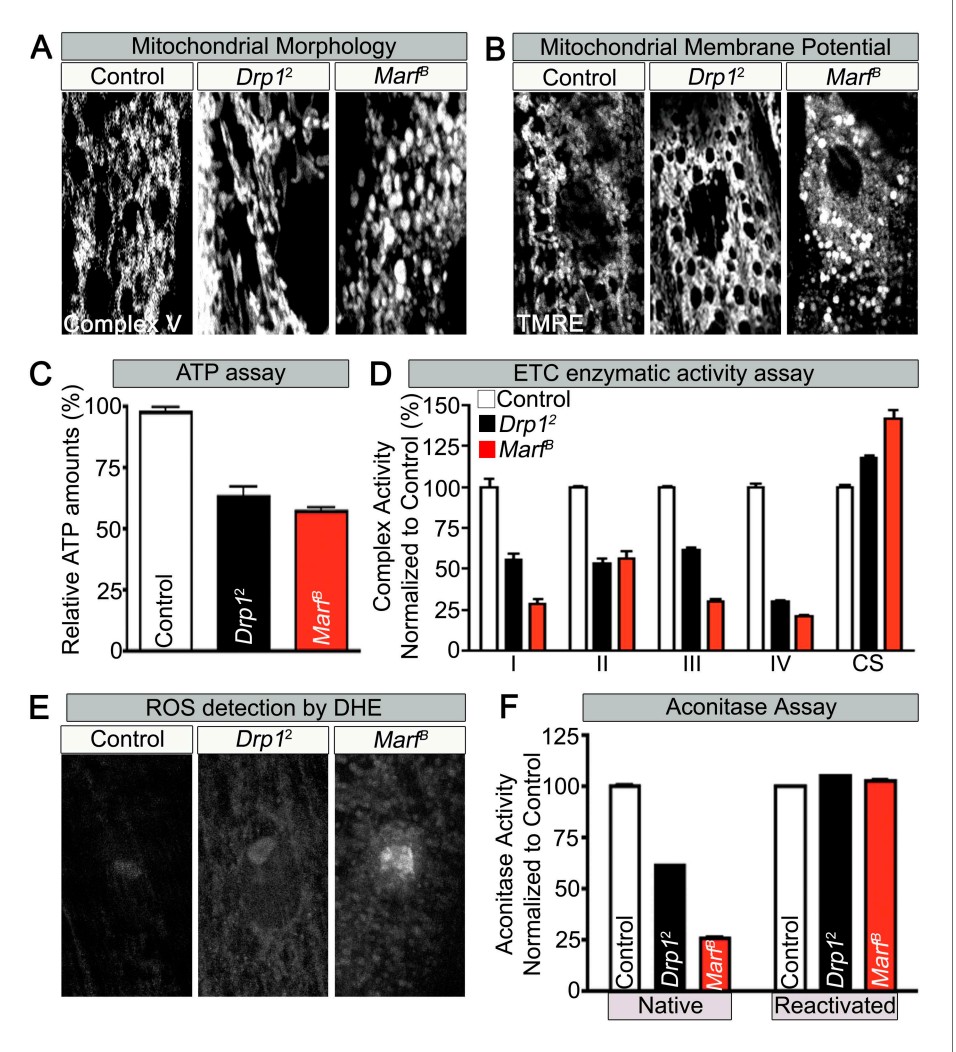

**Figure 2**. Mitochondrial morphology and function in *Marf* and *Drp1* mutants. (**A**) Mitochondrial morphology based on anti-Complex V antibody staining (Complex V) in larval muscles (Zoom in view around muscle nucleus). (**B**) Mitochondrial membrane potential as measured by the TMRE dye in larva muscle. (**C**) Relative ATP amounts. (**D**) Measurement of the enzymatic activity of electron transport chain (ETC) complexes (I–IV) from purified mitochondria from third instar larvae. All the ETC activities were normalized to citrate synthase (CS) activity of controls. (**E** and **F**) ROS is measured by two methods: (**E**) by DHE staining in larval muscles and (**F**) by measuring aconitase activity reduction from purified mitochondria. Reducing reagents reactivate native aconitase. Aconitase activities were normalized to controls. (**C**, **D** and **F**) error bars represent ± SEM.

The following source data are available for figure 2:

**Source data 1**. Lethal staging of Drp1 mutants.

**Source data 2**. Phenotypic comparison of Marf, Drp1 and Marf and Drp1 mutants.

mutants is reduced (***Figure 2B*** and ***Figure 2—source data 2***). Measurements of the activity of the Electron Chain Complexes (ETC I, II, III and IV) that pump protons across the mitochondrial inner membrane from the mitochondrial matrix to the inner membrane space to generate the MMP revealed that all ETC complex activities are similarly or more severely affected in *Marf* than *Drp1* mutants (***Figure 2D***). Furthermore, measurement of reactive oxygen species (ROS) by dihydroethidium (DHE) staining (***Shidara and Hollenbeck, 2010***) and mitochondrial aconitase assay (native activity of aconitase negatively correlates with ROS levels) (***Yan et al., 1997***) shows that *Marf* mutants are significantly more

severely affected than *Drp1* mutants (*Figure 2E,F* and *Figure 2—source data 2*). The ROS data is in agreement with the ETC data as loss of function of CI and CIII are considered the major drivers of increased ROS (*Koopman et al., 2013*). In summary, *Marf* and *Drp1* mutants exhibit dysfunctional mitochondria, but loss of *Marf* affects their function more severely.

Loss of one copy of *MFN2* in human causes a progressive and severe loss of function of neurons with long axons and affects motor neurons (MN) more severely than sensory neurons (*Zuchner et al., 2004*). To assess if mitochondria in MN are affected in larvae we expressed MitoGFP in MN using the *D42-Gal4* driver (*Pilling et al., 2006*). In the ventral nerve cord (VNC) of control larvae, MitoGFP mostly localizes to the neuropil (*Figure 3A*). *Marf* mutants show an obvious reduction in levels of mitochondria in the neuropil and the mitochondria mostly form clumps in the soma and the initial segments of axons (*Figure 3A*). In control MN, MitoGFP also labels numerous mitochondria in axons that innervate proximal (A3) and more distal (A5) segments (*Figure 3B*). In the axons of *Marf* mutants, fewer MitoGFP-marked mitochondria are observed in distal axons compared to controls (*Figure 3B*). These data show that loss of *Marf* impairs, but does not abolish, axonal mitochondrial transport (*Figure 3B*).

To assess the presence of mitochondria at NMJs, we counted MitoGFP positive puncta in boutons labeled by anti-Discs Large 1 (Dlg1 [*Parnas et al., 2001*]). While control NMJs contain numerous mitochondria per bouton, *Marf* boutons contain almost no mitochondria, even fewer than in *Drp1* mutants (*Figure 3C*, see Figure legend, [*Verstreken et al., 2005*]). However, unlike *Drp1* mutants, *Marf* mutant NMJs exhibit severe morphological defects (see below). Interestingly, we find no obvious labeling defects with the presynaptic active zone marker Bruchpilot (*Wagh et al., 2006*), endocytic markers such as α-Adaptin (*Gonzalez-Gaitan and Jackle, 1997*), Dap160 (*Roos and Kelly, 1998*), Endophilin (*Verstreken et al., 2002*), and Synaptojanin (*Verstreken et al., 2003*), or the postsynaptic Glutamate receptor IIA (*Qin et al., 2005*) in *Marf* mutants (*Figure 3—figure supplement 1*). Expression of Marf protein in MN using the *D42-Gal4* driver rescues the trafficking defect and restores the presence of mitochondria at the NMJ (*Figure 3*). However, it does not restore the morphological defects (*Figure 3C*), suggesting that Marf's function in mitochondrial trafficking is cell autonomous and that the defects in synapse morphology are cell non-autonomous.

Recently, mammalian MFN2 was shown to physically interact with MIRO2, an adaptor protein for motor proteins required for mitochondrial trafficking (*Misko et al., 2010*). *Drosophila miro* (*dmiro*) mutants are severely impaired in mitochondrial trafficking in the VNC (*Guo et al., 2005*). Indeed, RNAi knockdown of *dmiro* almost abolishes the presence of mitochondria in axons, a phenotype that is much more severe than what we observe in *Marf* mutants (data not shown). Moreover, loss of *dmiro* in *Marf* mutant MNs largely enhances the mitochondrial trafficking defect in the VNC and proximal axons (*Figure 3—figure supplement 2A,B*). This suggests that Marf cannot be the sole anchor that binds dMiro for mitochondrial trafficking.

## *Marf* is required to maintain synaptic transmission upon repetitive stimulation

Loss of mitochondria at NMJs in *Drp1* mutants was shown to affect synaptic transmission at high frequency stimulation (*Verstreken et al., 2005*). To gauge how loss of *Marf* affects synaptic transmission we performed electrophysiological recordings at the NMJs, using a transheterozygous *Marf^B^/Marf^E^* allelic combination in order to compare larvae of the same size since *Marf^B^* mutant are small in size. When stimulated at 0.2 Hz, *Marf* mutants do not exhibit any obvious defect in transmitter release based on excitatory junction potential (EJP) recordings (*Figure 4A*). Moreover, the amplitude of spontaneous release events or miniature EJPs (mEJPs) and quantal content are not altered in *Marf* mutants (*Figure 4A*). Hence, the average number of vesicles released in response to low frequency stimulations in *Marf* mutants is not different from *Marf* genomic-rescue controls. However, *Marf* mutant terminals are unable to properly sustain a 10 Hz stimulus for 10 min when compared to controls (*Figure 4B*) as the EJP amplitudes progressively decrease. A rundown of synaptic transmission is often observed in endocytic mutants such as *endophilin* and *synaptojanin* (*Verstreken et al., 2002*, *2003*; *Dickman et al., 2005*), *dap160* and *eps15* (*Koh et al., 2004*, *2007*), and *flower* (*Yao et al., 2009*). We therefore assessed if endocytosis is impaired and used FM1-43, a dye that reversibly binds membranes and is internalized into vesicles (*Verstreken et al., 2008*). Unlike *eps15* mutants that serve as a positive control, nerve stimulation at 60 mM K⁺ in the presence of FM1-43 effectively labels synaptic boutons in *Marf* mutants similar to controls (*Figure 4C,D*). Hence, vesicle endocytosis or evoked responses at 0.2 Hz are not affected in *Marf* mutants. These features are similar to *Drp1* mutants, suggesting that lack of

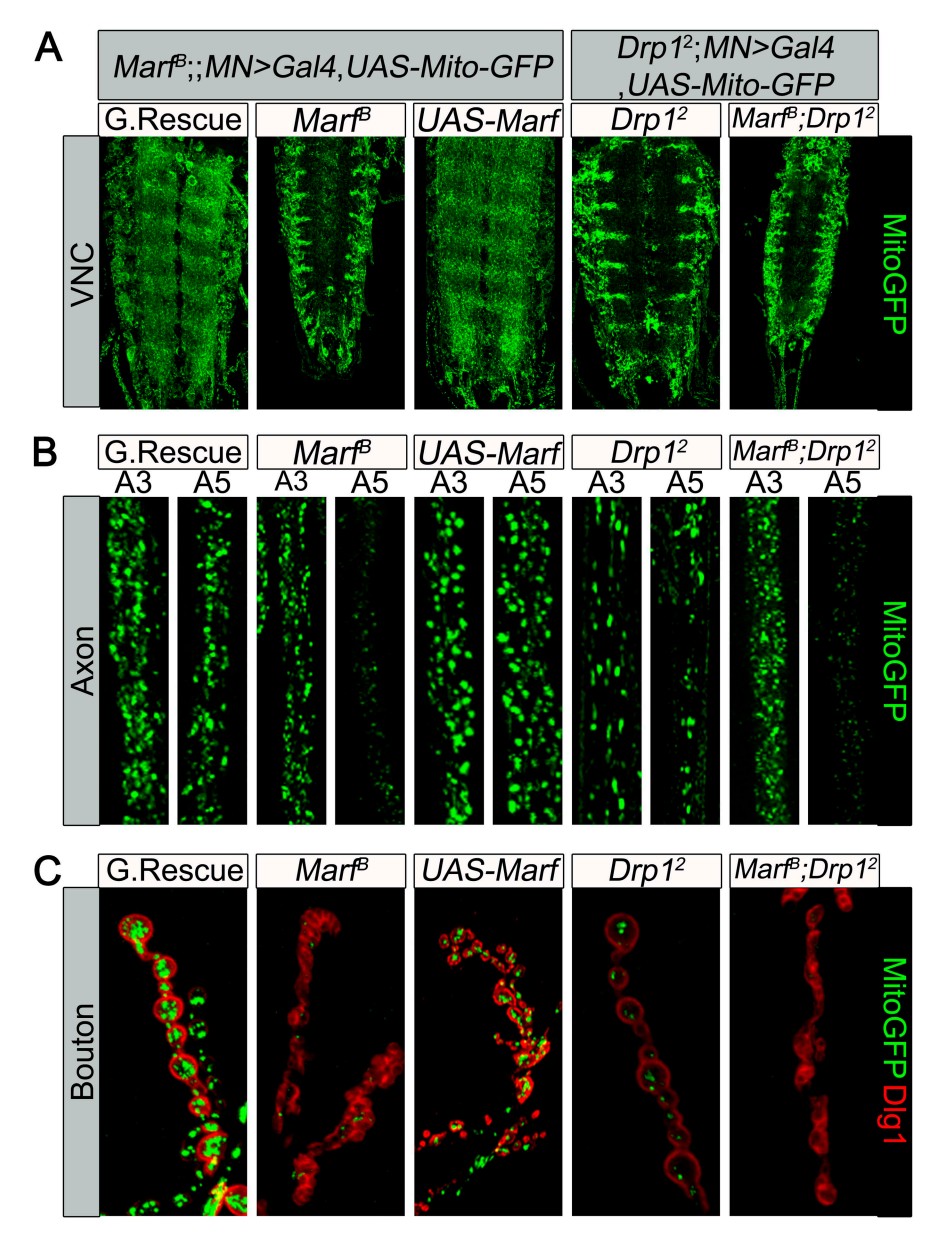

**Figure 3**. Mitochondrial trafficking defects in distal axons and boutons. Mutations and controls were crossed to a motor neuron driver (*D42-GAL4, UAS-MitoGFP*) to label neuronal mitochondria. (**A**) Ventral nerve cord (VNC): *Marf* and *Drp1* mutants exhibit clustered mitochondria in the soma. (**B**) Comparison of a proximal axonal segment in A3 and a distal segment in A5. Distal segments of A5 axons in *Marf* mutants contain many fewer mitochondria than proximal segments. (**C**) *Marf* mutants contain almost no mitochondria in boutons when co-stained with post-synaptic marker Discs Large 1 (Dlg1). Percentage of boutons with no mitochondria: Genomic rescue (0%), *Marf*$^B$ (89%), *UAS-Marf* (0%), *Drp1*$^2$ (36%) and *Marf*$^B$;*Drp1*$^2$ (95%).

The following figure supplements are available for figure 3:

**Figure supplement 1**. Pre-synaptic, endocytic and postsynaptic markers are present in Marf mutant boutons.

**Figure supplement 2**. Mitochondrial trafficking defect in Marf mutants cannot be rescued by motor neuron expression of human MFN1 or MFN2.

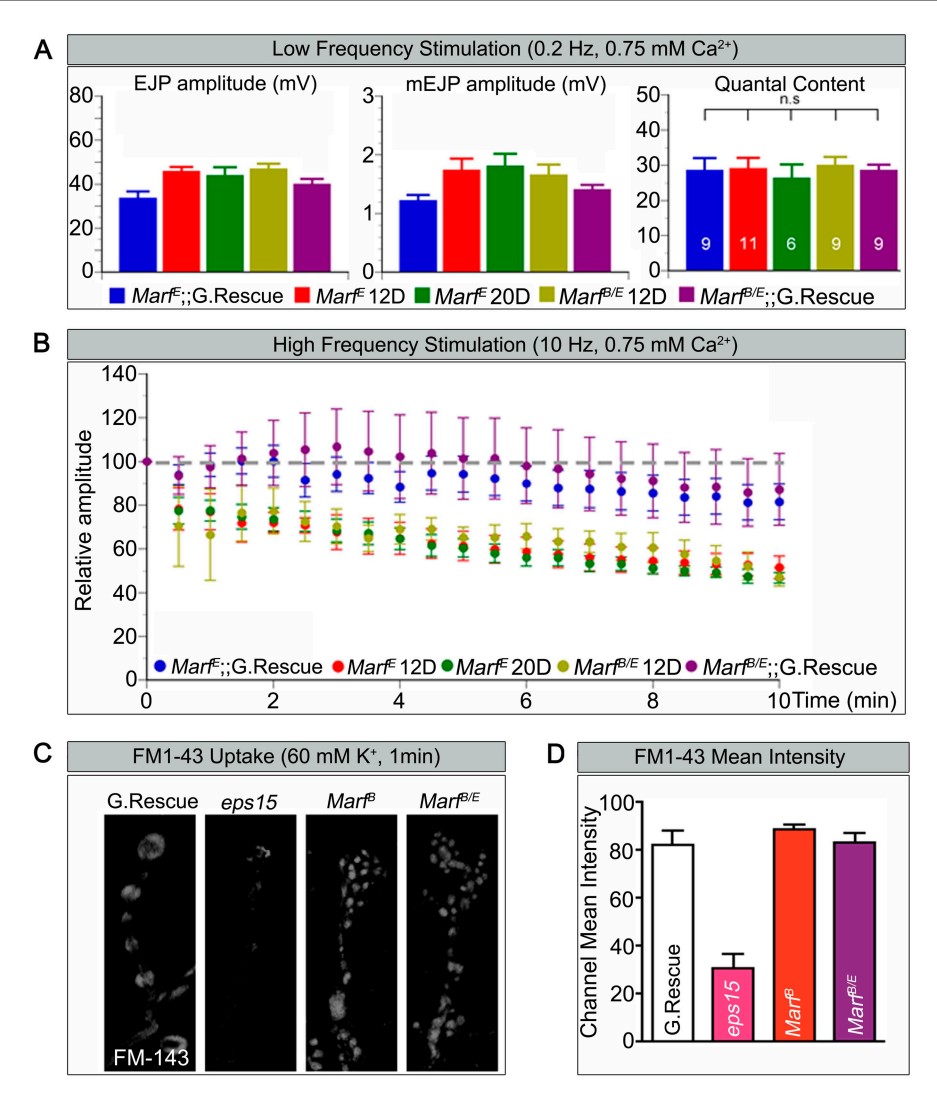

**Figure 4**. *Marf* is required to maintain synaptic transmission upon repetitive stimulation. (**A**) Excitatory Junctional Potentials (EJP) and miniature EJPs (mEJP) measured at 0.2 Hz in 0.75 mM $Ca^{2+}$ are similar in *Marf* mutants (day 12 or day 20 old larvae) and controls. Hence, quantal content in *Marf* mutants is also similar to controls (n = 6–11 larvae assayed). (**B**) Controls display facilitation whereas *Marf* mutants (day 12 or day 20 old larvae) show a rundown at 10 Hz in 0.75 mM $Ca^{2+}$. (**C**) Assessing endocytosis using FM-143 dye uptake at 60 mM [$K^+$] for 1 min shows no obvious differences between wild type controls and *Marf* mutants. (**D**) Quantification of FM-143 uptake. Error bars represent ± SEM.

mitochondria at synaptic terminals affect ATP levels required for vesicle mobilization at high frequency stimulation (*Verstreken et al., 2005*).

## *Marf* is required for proper NMJ development

A striking difference between *Marf* mutants and *Drp1* mutants is that loss of Marf severely affects NMJ morphology whereas loss of *Drp1* does not affect NMJ development (*Figure 3C*, *Figure 3—figure supplement 1* and *Figure 2—source data 2*). To visualize bouton morphology, we co-stained with Eps15, a presynaptic marker (*Koh et al., 2007*) and Dlg1, a postsynaptic marker (*Parnas et al., 2001*). *Marf* mutant displayed a severe reduction in average bouton size (*Figure 5A*) accompanied by an increase in clustering and numbers of boutons when compared to controls (*Figure 5A,C*). This NMJ phenotype can be rescued by a *Marf* genomic rescue construct as well as ubiquitous expression of a *Marf* cDNA (*Figure 5A,C*). An increase in bouton number and reduction in size is also observed by ubiquitous knockdown of *Marf* using RNAi (*Figure 5B,D* and *Figure 1—figure supplement 1C*).

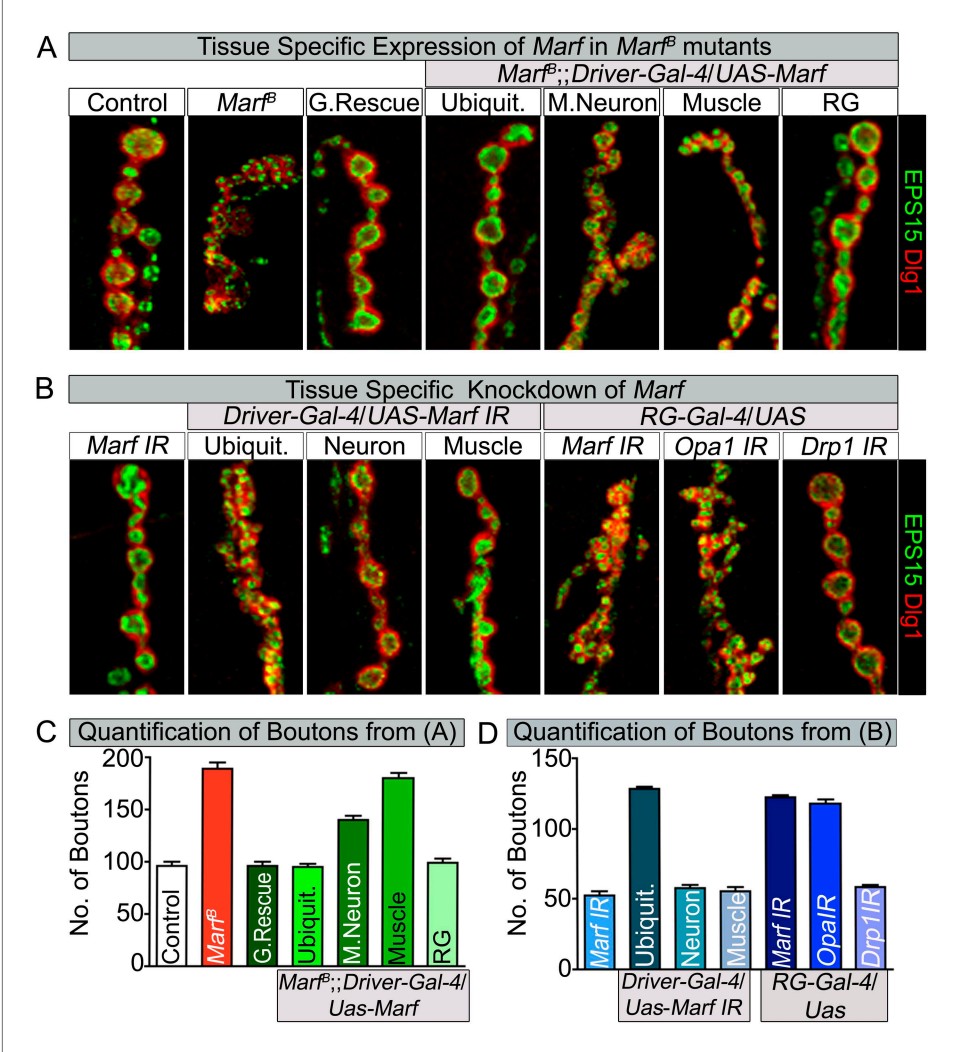

**Figure 5**. Loss of mitochondrial fusion but not fission in the ring gland results in altered bouton morphology. Third instar larvae NMJs from muscles 6/7 segments A3 were stained with pre-synaptic (EPS15) and post-synaptic (Dlg1) markers. (**A**) Ubiquitous (*Tubulin-Gal4*) or ring gland (RG, *Feb36-Gal4*) expression of *Marf* rescue bouton morphology in *Marf* mutants, while motor neuron (*D42-Gal4*) or muscle (*Mef-Gal4*) *Marf* expression did not. (**B**) Ubiquitous or RG specific knockdown of *Marf* or *Opa1* (**Poole et al., 2010**) phenocopy the bouton phenotype in *Marf* mutants while knockdown of *Drp1* (*Drp1 IR* knockdown of *Drp1* mRNA is 82% using ubiquitous driver Actin-Gal4) did not. (**C** and **D**) Quantification of bouton numbers from three independent experiments. Error bars represent ± SEM.

The following source data and figure supplements are available for figure 5:

**Source data 1**. Tissue specific *Gal4* screen to assess rescue of lethality and bouton morphology by Marf expression.
**Source data 2**. Tissue specific *Gal4* screen to assess lethality and alterations to bouton morphology by Marf knockdown.
**Figure supplement 1**. Ring gland drivers tissues specificity.

Since ubiquitous expression of the *Marf* cDNA rescues the NMJ morphology phenotype, we tested whether expression of Marf in MN, muscles or glial cells is able to rescue the phenotype. The NMJ phenotype is only partially rescued by Marf expression in MN (***Figure 5A,C***). Moreover, muscle, glial or MN and muscle expression of Marf does not alter the *Marf* mutant NMJ morphology (***Figure 5A,C*** and ***Figure 5—source data 1***). Consistent with these observations, RNAi knock down of *Marf* in MN, muscles, glia and MN and muscle does not affect bouton number or size at NMJs (***Figure 5D*** and

*Figure 5—source data 2*). This indicates that Marf expression is required in other cells than MN, muscles or glia.

## Mitochondrial fusion regulates NMJ morphology via a non-cell autonomous function in the ring glands

To assess which other tissue/cells contribute to the NMJ defects in *Marf* mutants, we tested specific RNAi knockdown of *Marf* using *Gal4* drivers that drive expression in different tissues including fat body, haemocytes, oenocytes, trachea or ring gland (RG) (*Figure 5—source data 2*). Knockdown of *Marf* with three independent *RG-Gal4* drivers resulted in a NMJ phenotype similar to that observed in *Marf* mutants or ubiquitous knockdown of *Marf* (*Figure 5B*, *Figure 5—source data 2* and *Figure 5—figure supplement 1*), clearly showing a non-cell autonomous requirement for Marf in RGs. In addition, while knockdown of *Marf* in neurons and RG resulted in pupal lethality, only knockdown of *Marf* in RG significantly lengthened the third instar larva stage (8–10 days) (*Figure 5—source data 2*). Finally, expression of Marf in the RG using two different RG drivers rescued the bouton phenotype of *Marf* mutants (*Figure 5A,C*, *Figure 5—source data 1* and *Figure 5—figure supplement 1*). Hence, Marf is required in RGs to regulate NMJ morphology in a cell non-autonomous manner.

Given that loss of *Drp1* does not cause obvious developmental defects at NMJs (*Figure 2—source data 2*, *Figure 3C* and *Figure 5B*) (*Drp1 IR* knockdown of *Drp1* mRNA is 82% using a ubiquitous driver *Actin-Gal4*), we tested whether loss of *Opa1*, another fusion protein (*Cipolat et al., 2004*; *Chen et al., 2005*), in RGs causes a bouton phenotype. A RG specific knockdown of *Opa1* (*Deng et al., 2008*; *Poole et al., 2010*) causes a very similar alteration in synaptic morphology as *Marf* knockdown (*Figure 5B*). Moreover, *Opa1* knockdown in RG also lengthens the larval stages and causes pupal lethality, similar to *Marf* knockdown (data not shown). Hence, both inner and outer mitochondrial fusion but not fission proteins alter bouton morphology and lengthen larval lifespan via RG, suggesting that the fusion proteins affect the same cell non-autonomous process.

RGs are responsible for production of hormones such as ecdysone (*Huang et al., 2008*) and juvenile hormone (*Di Cara and King-Jones, 2013*). These hormones regulate growth and differentiation of numerous tissues and control the proper timing of larval molts and metamorphosis (*Yamanaka et al., 2012*; *Di Cara and King-Jones, 2013*). Loss of production of ecdysone in RGs results in a lengthened larval stage ranging from 4 to 19 days (*McBrayer et al., 2007*, *Talamillo et al., 2008*; *Rewitz et al., 2009*). To determine if ecdysone production is affected we measured the levels of 20-hydroxyecdysone (20E) (*Porcheron et al., 1976*), in *Marf* mutants as well as animals with RG specific knockdown of *Marf*, *Opa* or *Drp1*. *Marf* mutants or knockdown of *Marf* and *Opa1* in RG exhibit severely reduced levels of 20E when compared to control or knockdown of *Drp1* in the RG or *Drp1* mutant alleles (*Figure 6A* and *Figure 2—source data 2*). Restoring expression of Marf in the RGs of *Marf* mutants partially restores the 20E levels (*Figure 6A*). Moreover, the feeding of 20E to third instar larvae with RG specific knockdown of *Marf* rescued both the pupal lethality and NMJ morphology phenotype (Data not shown and *Figure 6—figure supplement 1A*). In summary, Marf and Opa1 but not Drp1 affect ecdysone production in the RG.

### *Marf* is required for lipid droplet formation in RG

The production of ecdysone (steroid hormones) involves many steps following uptake of cholesterol. *Drosophila* lacks several biosynthetic enzymes for de novo cholesterol synthesis and depends on cholesterol uptake from the food (*Clark and Block, 1959*). In the RG, cholesterol is processed into 'free-cholesterol (FC)' in the ER (*Miller, 2013*). It is then transported into the mitochondrial inner matrix for processing by at least two cytochrome p450 enzymes (encoded by *disembodied* [*Chavez et al., 2000*] and *shadow* [*Warren et al., 2002*] in *Drosophila*) and finally secreted from the RG into the hemolymph (*Gilbert, 2004*). Because steroid hormones cannot be stored during *Drosophila* larva development, FC is stored in the form of cholesterol esters in lipid droplets (LDs) until there is a burst of ecdysone synthesis (*Talamillo et al., 2008*; *Miller, 2013*). This process of cholesterol ester storage and steroid synthesis is highly conserved from flies to mammals.

To assess cholesterol ester storage in LDs in RGs of wandering third instar larva, we first stained LDs with Nile Red, which marks neutral lipids that comprise LDs (*Greenspan et al., 1985*). This larval stage precedes the large burst of ecdysone that occurs at the larval–pupal transition (*Yamanaka et al., 2012*). Interestingly, the numbers of LDs are severely reduced in *Marf* mutants as well as in *Marf* knockdown in RGs (*Figure 6B,C*). Moreover, RG expression of *Marf* rescues the LD phenotype and even

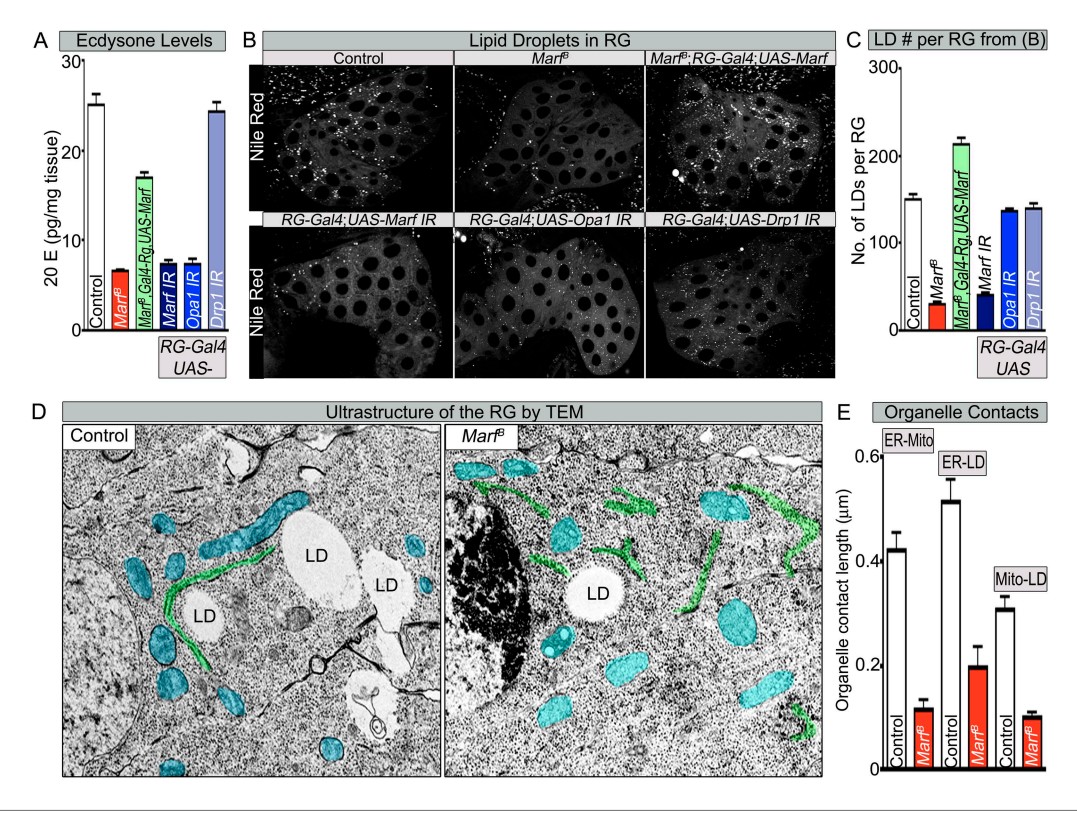

**Figure 6**. Both Marf and Opa1 regulate ecdysone synthesis in the ring gland, but only Marf promotes lipid droplet formation. (**A**) Both loss of *Marf* and *Opa1* in the RG have reduced 20-hydroxyecdysone (20E) levels when compared to loss of *Drp1* and controls. 20E levels are determined and normalized by weight. (**B**) Only loss of *Marf* in the RG results in reduced lipid droplets (LDs) when stained by Nile Red compared to loss of *Opa1* or *Drp1*. (**C**) Quantification of LDs in the ring gland (RG) from three independent experiments. (**D**) TEM sections of RG were the ER is labeled in green, mitochondria in blue and lipid droplets are labeled 'LD'. *Marf* mutants display increased ER fragmentation and reduced numbers of LDs when compared to *Marf*-genomic rescue control animals. (**E**) *Marf* mutants have reduced contact length between mitochondria and ER, ER and LD, and mitochondria and LD when compared to controls. Error bars represent ± SEM.

The following figure supplement is available for figure 6:

**Figure supplement 1**. Feeding of 20E rescues the NMJ morphology of RG specific knockdown of Marf.

increases the LDs numbers above control in *Marf* mutants, suggesting that Marf is necessary and sufficient for LD formation (**Figure 6B,C**). Interestingly, RG knockdown of *Opa1* does not affect LD number (**Figure 6B,C**), suggesting that Marf and Opa1 have different roles in the RG. Our findings indicate that Marf plays a unique role in LD synthesis in RG and that it affects cholesterol ester storage. Loss of *Opa1* on the other hand does not affect LD storage but like loss of *Marf*, impairs 20E production. Finally, loss of *Drp1* or RG expression of Drp1 does not affect LD synthesis, nor does it affect 20E production (**Figure 6A–C**, **Figure 2—source data 2** and **Figure 6—figure supplement 1B,C**). Taken together, the three mitochondrial GTPases have different roles in LD dynamics and ecdysone synthesis.

LD are generated from the ER through budding of the outer leaflet of the ER membrane (**Walther and Farese, 2012**). A physical link between the ER, LDs and mitochondria are often observed as these organelles collaborate to orchestrate numerous metabolic processes such as cholesterol transport and steroid synthesis (**Issop et al., 2012**; **English and Voeltz, 2013**). Indeed, human MFN2 has been shown to tether the mitochondria to the ER (**de Brito and Scorrano, 2008**). To assess the ultrastructural features of ER, LDs, and mitochondria in RGs, we performed TEM in RG. As shown in **Figure 6D**, *Marf* mutants exhibit a fragmented ER, reduced number of LD, and morphologically altered mitochondria when compared to controls. The contacts between the mitochondria and the ER, the ER and LD, as

well as mitochondria and LD, are all severely reduced in *Marf* mutant RG (*Figure 6D,E*). This suggests that Marf promotes cholesterol ester storage in LDs possibly through inter-organelle connections.

### *Marf* integrates the functions of human MFN1 and MFN2

Human MFN2 tethers mitochondria to the ER (*de Brito and Scorrano, 2008*) but this has not been documented for MFN1. Similarly, loss of *MFN2* leads to ER stress (*Ngoh et al., 2012*; *Sebastian et al., 2012*; *Munoz et al., 2013*) but a role for MFN1 in ER function has not been reported. If *Drosophila* Marf mediates connections of mitochondria to ER and if this activity is required for ecdysone synthesis, expression of human *MFN2* (*Dorn et al., 2011*) in the RG may rescue the loss of LDs, alleviate the bouton morphology defects and restore 20E levels in *Marf* mutants. We find that RG specific expression of human *MFN2* restores the proper number of LD levels and organelle contacts in *Marf* mutants whereas expression of human *MFN1* (*Dorn et al., 2011*) does not (*Figure 7A,C* and *Figure 7—figure supplement 1*), indicating that MFN2 specifically can rescue the defect in LD synthesis. However, RG expression of human *MFN2* did not rescue the bouton phenotype of *Marf* mutants (*Figure 7B,D*). Moreover, ubiquitous expression of *MFN1* or *MFN2* alone (*Daughterless-Gal4* and *Tubulin-GAL4*) does not rescue the lethality (*Figure 1—figure supplement 1B*), mitochondrial morphology (*Figure 7—figure supplement 2*), mitochondrial trafficking to synapses (*Figure 3—figure supplement 2*), 20E levels, and the NMJ phenotypes (*Figure 7*), whereas ubiquitous co-expression of both *MFN1* and *MFN2* rescued all phenotypes (*Figure 1—figure supplement 1B* and *Figure 7*). These data indicate that MFN1 and MFN2 play non-redundant roles and have complementary functions that are integrated into a single protein in *Drosophila* Marf.

## Discussion

How does loss of fission or fusion affect mitochondrial function? In the absence of fusion mixing of mitochondrial DNA and proteins may be severely impaired. Given that mitochondrial proteins are in an environment rich in oxygen radicals, lack of fusion may cause more damage than when fission is impaired (*Chan, 2012*). Simply stated, loss of fusion proteins like Marf, MFN1 or MFN2 may cause more severe phenotypes than the loss of a fission protein like Drp1. Moreover, proteins like Marf and Drp1 may perform other functions that are not directly related to fusion or fission, and hence affect other processes. Based on a careful phenotypic comparison of loss of *Marf* and *Drp1* in *Drosophila* we find many similarities and differences.

*Marf* mutants display small mitochondria whereas *Drp1* mutants exhibit large fused mitochondria. Interestingly, both mutants accumulate mitochondria in the cell body of the neurons and the proximal axonal segments (*Figure 3A*). In *Drp1* mutants, the mitochondria seem to be severely elongated in axons where they fail to reach the NMJs, as previously described (*Verstreken et al., 2005*). The impairment in axonal transport is thought to be due to the fact that the mitochondria are hyperfused and cannot easily be transported. Indeed, loss of *Marf* in *Drp1* mutants can restore mitochondrial trafficking proximally but distal axonal trafficking is still impaired (*Figure 3B*). In *Marf* mutants, even though mitochondria are small and can enter the axons, the numbers of mitochondria that travel distally toward the NMJs are dramatically reduced (*Figure 3B*). Hence, loss of *Marf* impairs mitochondrial trafficking and longer axons are more severely affected than shorter axons. Since longer axons are more severely affected in CMT2A patients (*Scherer, 2011*), defects in mitochondrial trafficking may be at the root of some of the phenotypes associated with the disease.

Mfn2 has been implicated in axonal transport via binding to Miro2. Indeed, knockdown of *MIRO2* in cultured vertebrate neurons affects mitochondrial transport in an identical fashion as loss of *MFN2* (*Misko et al., 2010*). However, the severity of mitochondrial transport that we observe in *Marf* mutants is much less pronounced than what has been described in *dmiro* mutants (*Guo et al., 2005*) and what we observe when *dmiro* is lost. Moreover, removal of d*miro* in *Marf* mutants dramatically enhances the Marf phenotype and almost abolishes axonal localization of mitochondria (*Figure 3—figure supplement 2*), arguing that Marf cannot be solely responsible for mitochondrial transport in *Drosophila*.

A comparison of the presence of mitochondria at NMJ synapses shows that *Marf* mutants have fewer mitochondria than *Drp1* mutants (*Figure 3C*). Moreover, *Marf* mutants but not *Drp1* mutants display a severe increase in small clustered boutons (*Figure 2—source data 2*, *Figures 3C and 5*). The small and clustered boutons have also been observed in other mutants like *endophilin* (*Dickman et al., 2006*), *synaptojanin* (*Dickman et al., 2006*), *eps15* (*Koh et al., 2007*), *dap 160* (*Koh et al., 2004*), *flower* (*Yao et al., 2009*) and *dmiro* (*Guo et al., 2005*). However, unlike in *Marf* mutants, the bouton

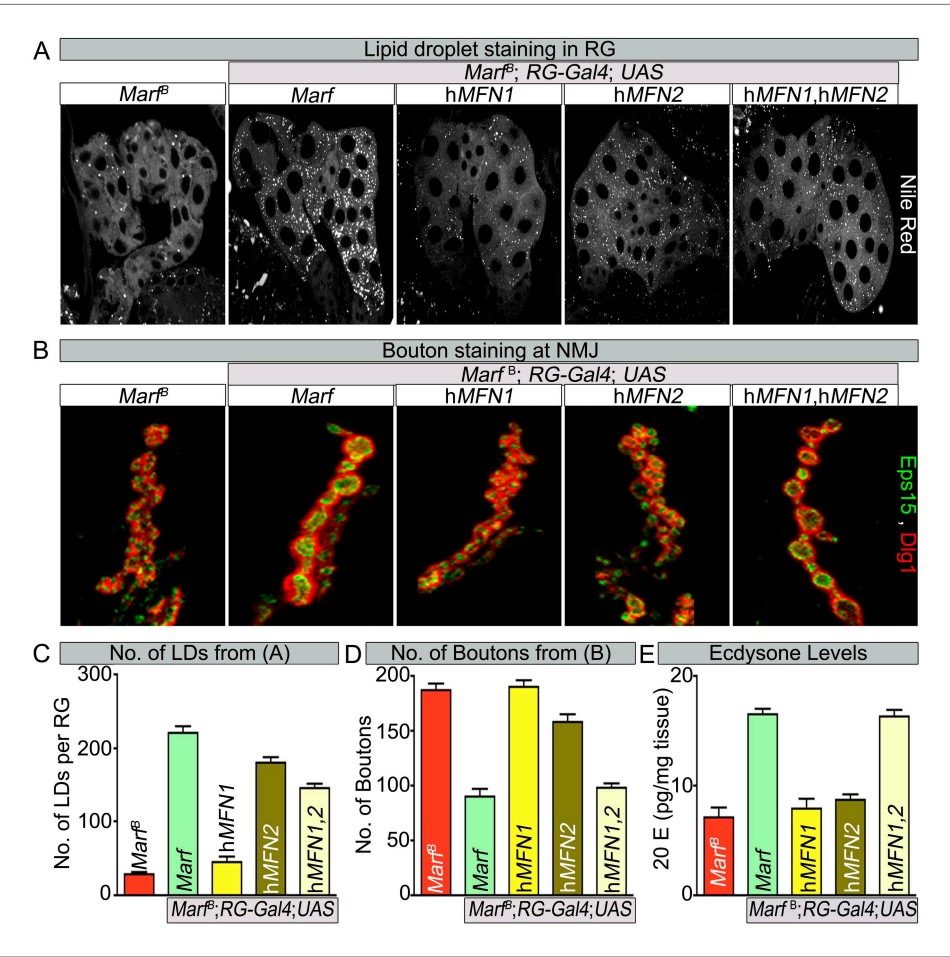

**Figure 7**. Human MFN2 restores LD numbers but both human MFN1 and MFN2 are required for steroid-hormone production in the ring glands. (**A**) Rescue of lipid droplets numbers stained by Nile Red in *Marf* ring glands (RG) by *MFN2* and *MFN1*/*MFN2* co-expression, but not *MFN1*. (**B**) Rescue of *Marf* bouton morphology by expressing *MFN1*/*MFN2* in RGs (*Feb36-Gal4*). Expression of *MFN1* or *MFN2* alone does not rescue the phenotype. (**C–E**) Quantification in control and *Marf* mutants for: (**C**) LDs (**D**) Boutons and (**E**) Ecdysone (20E levels) as described in *Figures 5 and 6*. Error bars represent ± SEM.

The following figure supplements are available for figure 7:

**Figure supplement 1**. RG expression of human MFN2 restores organelle contact lengths in Marf mutants.

**Figure supplement 2**. Muscle expression of either human MFN2 or MFN1 does not fully restores mitochondrial morphology in Marf mutants.

phenotypes are fully rescued by neuronal expression of the cognate protein within MN in the above mentioned mutants. Moreover, knockdown of *Marf* in neuron, muscle or glia does not recapitulate the bouton phenotype observe in Marf mutants (*Figure 5B* and *Figure 5—source data 2*), suggesting a unique cell non-autonomous requirement of Marf for proper NMJ morphology.

*Marf* mutants exhibit two obvious phenotypes at NMJs: a severe depletion of mitochondria and a doubling of the number of boutons combined with a severe reduction in size whereas *Drp1* mutants only exhibit a severe reduction in mitochondria. However, our electrophysiological studies show that loss of *Marf* does not affect basal synaptic transmission (*Figure 4*) similar to what is observed in *Drp1* mutants (*Verstreken et al., 2005*). Both respond similarly to wild type NMJs when stimulated at 0.2 Hz and both show a progressive run down at 10 Hz when compared to controls. Moreover, endocytosis using FM1-43 and 60 mM K⁺ is not impaired in *Marf* and *Drp1* mutants, suggesting a defect in reserve

pool mobilization in both mutants (*Verstreken et al., 2005*, *2008*). The data also show that the bouton defects observed in *Marf* mutants do not contribute to the run down in synaptic transmission since *Drp1* boutons are normal in number and size yet also have a run down in synaptic transmission (*Figure 2—source data 2*, *Figures 3 and 4*; [*Verstreken et al., 2005*]).

Loss of *Marf* in RG recapitulates the bouton phenotype observed in *Marf* mutants and expression of *Marf* in RG fully rescues this phenotype (*Figure 5* and *Figure 5—source data 1*). Interestingly, both Marf and Opa1 are required for steroid hormone production and both lead to extended larval lifespan when knocked down in the RG only (8–10 days), whereas *Drp1* mutations do not affect steroid hormone synthesis. Reduction of ecdysone production by knockdown of the prothoracicotropic hormone receptor (torso) in the RG also leads to an extended larval lifespan (9 days) (*Rewitz et al., 2009*) and an increased growth of NMJs (*Miller et al., 2012*). Interestingly, knockdown of *Drosophila* SUMO (*dsmt3*) in RG lead to a defect in cholesterol import in the RG, reduced 20E levels and an extended larval lifespan (19 days) (*Talamillo et al., 2008*). Hence, the severe reduction in ecdysone synthesis in *Marf* mutant RG underlies the prolonged larva stages and NMJ morphological defects.

The reduction in the number of LDs in RGs when Marf is lost suggests that these RGs are unable to store cholesterol (*Figure 6B,C*). This storage of cholesterol esters probably permits the RG to produce large amounts of ecdysone when needed, especially at the larval stage and larval to pupal transitions. Cholesterol storage and steroid hormone biosynthesis requires both the ER and mitochondria in vertebrates (*Miller, 2013*) but loss of *MFN1* or *MFN2* have not been shown to affect LD synthesis. Defects of anchoring mitochondria to the ER and LDs in *Marf* RGs argue that these defects lead to the loss of LD and production of ecdysone (*Figure 6*). In agreement with this hypothesis, expression of human MFN2, which tethers ER to mitochondria (*de Brito and Scorrano, 2008*), in *Marf* mutants restores LD synthesis and organelle contacts (*Figure 7A*, *Figure 7C* and *Figure 7—figure supplement 1*). Moreover, expression of human *MFN2* in RNAi mediated *Marf* knockdown in neurons and muscles rescues ER morphology and stress (*Debattisti et al., 2014*). However, *MFN2* expression alone in *Marf* mutant RG did not restore ecydsone synthesis (*Figure 7E*), arguing that there are other mitochondrial defects associated with the loss of *Marf* (*Figure 8*).

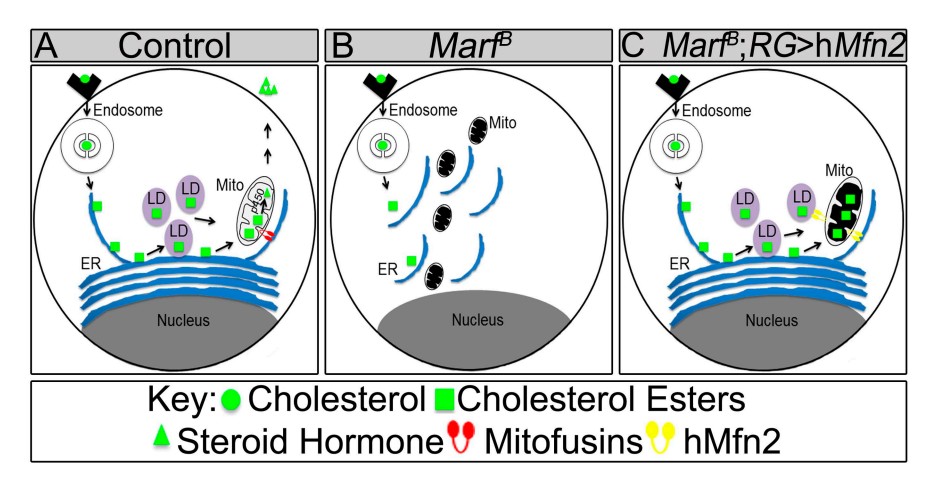

**Figure 8**. Model of Marf dual function in steroid synthesis in the ring glands. (**A**) In wild type ring glands (RG), cholesterol must enter the cell first. Then, cholesterol undergoes a series of modifications in endosomes and along the ER to become free-cholesterol. Then, free-cholesterol is transferred into the mitochondrial inner matrix, where it is processed from free-cholesterol to steroid hormone by p450 enzymes. The steroid hormone is then secreted. As *Drosophila* larva develops it stores cholesterol in the form of cholesterol ester in lipid droplets (LDs) in order to accumulate a reserve of substrate so it can generate bursts of steroid hormone when needed. These LDs require the ER for synthesis. (**B**) In *Marf* mutants, the ER is fragmented and LD formation is severely reduced. (**C**) RG-specific expression of *MFN2* in *Marf* mutant restores LD numbers but does not rescue hormone synthesis, suggesting that Marf has a second function within the mitochondria.

Our data show that co-expression of human *MFN1* and *MFN2* fully rescue the observed phenotypes in *Marf* mutants (*Figure 7*). Although RG-specific expression of MFN1 in *Marf* mutants did not restore LD numbers or organelle contacts (*Figure 7—figure supplement 1*), MFN1 is still necessary for ecdysone synthesis together with MFN2, suggesting a role downstream of cholesterol ester storage for both proteins (*Figure 8*). Moreover, knockdown of *Opa1* in RG did not alter LD numbers but causes reduced 20E levels and aberrant NMJs (*Figure 6*). Opa1 resides within the inner mitochondrial membrane, suggesting its role in ecdysone synthesis is within the mitochondria. Ecdysone synthesis within the mitochondria requires two cytochrome p450 enzymes encoded by *disembodied* (*Chavez et al., 2000*) and *shadow* (*Warren et al., 2002*). Hence, it is likely that impairment in fusion but not fission affects the function of these enzymes (*Figure 8*).

Opa1 and MFN2 but not Drp1 have been implicated in vertebrate steroidogenesis (*Issop et al., 2012*). Interestingly, in placental trophoblast cells (BeWO) in culture the loss of *OPA-1* promotes progesterone production by 70% whereas loss of *MFN2* has been reported to lead to a 20% decrease in progesterone production (*Wasilewski et al., 2012*). In contrast, testosterone production in MA-10 Leydig cells was unaffected by loss of *OPA1* (*Rone et al., 2012*) whereas loss of *MFN2* did affect testosterone production by 40% in MA-10 Leydig cells (*Duarte et al., 2012*). Hence, in both vertebrate endocrine cells, loss of *MFN2* or *OPA-1* affected steroids very differently as we observe very similar phenotypes associated with the loss of either protein. Our study also suggests that MFN2 functions upstream of cholesterol entry into the mitochondria at the cholesterol storage stage, since MFN2 restores LD synthesis in *Drosophila* RG. However, rescuing LD production is not sufficient to restore ecdysone synthesis, suggesting a secondary defect (*Figure 8C*). In summary, our data indicate that MFN1 and MFN2 have separate functions in vivo that are integrated in a single protein in fly Marf.

## Materials and methods

### Fly Strains, maintenance of flies and larvae

Flies were obtained from the Bloomington Drosophila Stock Center at Indiana University (BDSC) unless otherwise noted. All flies were kept in standard media and stocks were maintained at room temperature (21–23°C). For all the larvae experiments described, flies were allowed to lay embryos for 48 hr on grape juice plates with yeast paste. Hemizygous mutant larvae and wild type controls were isolated via GFP selection at the first instar phase and transferred to standard fly food for the duration of their development.

The following stocks were used in this study:

1. $y^1$ $w^*$ P{neoFRT}19A
2. $y^1$ $w^*$ Marf$^{A,B,C,E,F,G\ or\ H}$ P{neoFRT}19A/FM7c,Kr-Gal4 UAS-GFP,sn$^+$
3. yw eyFLP GMR-LacZ; y$^+$; Drp1$^2$ FRT40A/CyO, Kr-Gal4 UAS-GFP
4. cl(1) P{neoFRT}19A/Dp(1;Y)y+ v+ ey-FLP
5. $y^1$w$^{118}$ ey-FLP; Drp1$^2$ FRT40A/CyO, Kr-Gal4 UAS-GFP
6. $y^1$ $w^*$ Marf$^{B\ or\ E}$ P{neoFRT}19A/FM7c,Kr-Gal4 UAS-GFP;; Genomic Marf-HA/TM6B,Tb+
7. $y^1$ $w^*$ Marf$^B$ P{neoFRT}19A/FM7c,Kr-Gal4 UAS-GFP;; UAS-MarfHA/TM6B,Tb
8. y w;; D42-Gal4, UAS-mito-HA-GFP, e/TM6B,Tb
9. y w; Drp1$^2$ FRT40A/CyO, Kr-Gal4 UAS-GFP; D42-Gal4, UAS-mito-HA-GFP, e/TM6B,Tb
10. $y^1$ $w^*$ Marf$^B$ P{neoFRT}19A/FM7c,Kr-Gal4 UAS-GFP; Drp1$^2$ FRT40A/CyO, Kr-Gal4 UAS-GFP
11. y w; Df(2L)burK1, eps15[e75]/Cyo; twi-Gal4 UAS-2xEGFP
12. $y^1$ $w^*$ Marf$^B$ P{neoFRT}19A/FM7c,Kr-Gal4 UAS-GFP;; Tub-Gal4/TM6B,Tb
13. $y^1$ $w^*$ Marf$^B$ P{neoFRT}19A/FM7c,Kr-Gal4 UAS-GFP; DA-Gal4
14. $y^1$ $w^*$ Marf$^B$ P{neoFRT}19A/FM7c,Kr-Gal4 UAS-GFP;; Mef-Gal4/TM6B,Tb
15. $y^1$ $w^*$ Marf$^B$ P{neoFRT}19A/FM7c,Kr-Gal4 UAS-GFP; Feb36-Gal4/CyO, Kr-Gal4 UAS-GFP
16. $y^1$ $w^*$ Marf$^B$ P{neoFRT}19A/FM7c,Kr-Gal4 UAS-GFP;; Mai60-Gal4/TM6B,Tb
17. y w;; UAS-Marf IR/T(2;3)TSTL,Cyo:TM6b,Tb
18. y w;; UAS-Drp1 IR/T(2;3)TSTL,Cyo:TM6b,Tb
19. y w;; UAS-dmiro IR/T(2;3)TSTL,Cyo:TM6b,Tb
20. $y^1$ $w^*$ Marf$^B$ P{neoFRT}19A/FM7c,Kr-Gal4 UAS-GFP;; UAS-MFN1/TM6B,Tb
21. $y^1$ $w^*$ Marf $^{alleles}$ P{neoFRT}19A/FM7c,Kr-Gal4 UAS-GFP;; UAS-MFN2/TM6B,Tb
22. yw eyFLP GMR-LacZ; y$^+$; Drp1$^1$ FRT40A/CyO, Kr-Gal4 UAS-GFP
23. Drp1$^{[T26]}$ cn bw sp/CyO, Kr-Gal4 UAS-GFP

24. *y; Drp1[KG03815]/CyO; ry*
25. *w; UAS-Drp1/TM6C, Sb Tb*
26. *Gal4* BDSC fly lines listed on *Figure 5—figure supplement 1*

## Screen and mapping of *Marf*

*y,w,P{neoFRT}19A[isogenized]* (iso) male flies were treated with low concentration of ethylmethanesulfonate to induce mutations, and mutant alleles which showed ERG defects were isolated as described (*Xiong et al., 2012*; *Zhang et al., 2013*; *Yamamoto et al., 2014*).

For mapping of the *Marf* group, male large duplications (~1–2 Mb) covering the X chromosome (*Haelterman et al., 2014*) were crossed with female *y,w mut*,P{neoFRT}19A[isogenized]* flies that were balanced with *FM7c,Kr-GAL4,UAS-GFP(Kr > GFP)*. For the *Marf* group, the lethality of all alleles were rescued by *Dp(1;Y)dx[+]5,y[+]/C(1)M5* (4C11;6D8 + 1A1;1B4). *Marf* alleles complemented with all the available deficiencies covered by *Dp(1;Y)dx[+]5,y[+]/C(1)M5* except *Df(1)Exel6239* (*Parks et al., 2004*; *Cook et al., 2012*). We then performed Sanger sequencing for genes located to this region and identified mutations in *Marf*.

## *Marf* genomic and cDNA constructs

A 6.1 kb genomic rescue fragment (X: 6259600…6265700, *Drosophila melanogaster* Release 5.7) was amplified using PCR from the P[acman] CH322-102K19 (*Venken et al., 2009*). This DNA fragment was then subcloned into the *HindIII* and *KpnI* sites of the *P* element transformation vector *P{CaSpeR-4-HA}* (*Yao et al., 2009*) and sequenced. For cDNA constructs, the CDS of *Marf* was retrieved from cDNA clones RE04414 (*Stapleton et al., 2002*), respectively, and subcloned into *pUAST-HA* vector (*Ohyama et al., 2007*) using *NotI* and *XbaI* sites. Cloning and DNA purification were performed based on standard protocols. All constructs were sequenced before injection.

## Generation of transgenic miRNAi lines for *Drosophila Marf*, *Drp1* and *dmiro*

As previously described in *Yao et al. (2008)*, we chose the 22 nucleotides of the coding sequence of *Marf*, *Drp1*, or *dmiro* as target sequences listed in lowercase and bold in the sequences shown below. In oligo-1, the third nucleotide from 3' end was changed to C. To synthesize essential backbone for miRNAi production, four long primers were designed. The first PCR product was generated by oligo-1 and -2. With the first PCR template, the final construct was generated by using common oligo-3 and -4 then digested with EcoRI and NotI and cloned into the pUAST transformation vector.

### *Marf-oligo-1*
GGCAGCTTACTTAAACTTAATCACAGCCTTTAATGTt**aaatgtggtgaacatcaaCca** TAAGTTAATATACCATATC

### *Marf-oligo2*
AATAATGATGTTAGGCACTTTAGGTAC**taaatgtggtgaacatcaaaca**TAGATATGGTATATTAACTTATGGT

### *Drp1-oligo1*
GGCAGCTTACTTAAACTTAATCACAGCCTTTAATGT**caacgcacgtggtcaacctCac**TAAGTTAATATACCATATC

### *Drp1-oligo2*
AATAATGATGTTAGGCACTTTAGGTAC**caacgcacgtggtcaacctaac**TAGATATGGTATATTAACTTAGTGA

### *Miro-oligo1*
GGCAGCTTACTTAAACTTAATCACAGCCTTTAATGT**gaatgtggttaattgcatcCac**TAAGTTAATATACCATATC

### *Miro-oligo2*
AATAATGATGTTAGGCACTTTAGGTAC**gaatgtggttaattgcatcaac**TAGATATGGTATATTAACTTAGTGG

### Common oligos
#### Oligo-3
GGCGAATTCATGTTTAAAGTCCACAACTCATCAAGGAAAATGAAAGTCAAAGTTGGCAGCTTACTTAAACTTAATCA

Oligo-4

GGCGCGGCCGCATCCAAAACGGCATGGTTATTCGTGTGCCAAAAAAAAAAAAAATTAAATAA
TGATGTTAGGCACTT

## Electroretinograms

For ERG recording, *y w *mut (lethal) FRT19A/FM7c, Kr-Gal4, UAS-GFP* flies were crossed to *y w P{w+} cl(1) FRT19A/Dp(1;Y)y+; eyFLP* or *y w; Drp1² FRT40A/CyO* crossed to *y w, eyFLP; Drp1² FRT40A/CyO* to generate flies with mutant clones in the eyes and ERGs were performed as previously described (*Ly et al., 2008*). Briefly, adult flies were glued to glass slides. A recording probe was placed on the surface of the eye, and a reference probe was inserted in the thorax. A 1-s flash of white light was given, and the response was recorded and analyzed by the AXON™-pCLAMP8 software.

## Transmission electron microscopy (TEM) of laminas and ring glands

TEM of photoreceptor terminals (*Verstreken et al., 2003*) and ring glands (*Bellen and Budnik, 2000*) was performed as described. TEM of photoreceptor terminals and ring glands were done using a Ted Pella Bio Wave processing microwave with vacuum attachments. Briefly, fly heads or third instar larva were dissected and fixed at 4°C in 4% paraformaldehyde, 2% glutaraldehyde, 0.1 M sodium cacodylate, and 0.005% $CaCl_2$ (PH 7.2) overnight, post-fixed in 1% $OsO_4$, dehydrated in ethanol and propylene oxide, and then embedded in Embed-812 resin (Electron Microscopy Sciences, Hatfield, PA). Photoreceptors or ring glands were then sectioned and stained in 4% uranyl acetate and 2.5% lead nitrate. TEM images of PR sections were taken using a JEOL JEM 1010 transmission electron microscope with an AMT XR-16 mid-mount 16 mega-pixel digital camera.

## Mitochondria functional assays for *Marf* and *Drp1* mutants

Staining of mitochondria membrane potential (MMP) by Tetramethylrhodamine ethyl ester (TMRE; Molecular Probes, Life Technologies, Grand Island, NY) and ROS by dihydroethidium dye (DHE; Sigma, St. Louis, MO) in live muscles, larvae were prepared and stained as described in *Shidara and Hollenbeck (2010)*. Live images were acquired using a 40× water immersion lens and a Zeiss LSM510 confocal microscope. ATP levels in larvae was determined as described (*Park et al., 2006*) using a kit (Invitrogen, Life Technologies, Grand Island, NY). Quantification of ETC enzymatic activity assay and aconitase assay were performed on isolated mitochondria extracted as previously described (*Graham et al., 2010*; *Zhang et al., 2013*). Enzymatic activity assays were performed as previously described (*Emptage et al., 1983*; *Das et al., 2001*; *Graham et al., 2010*; *Zhang et al., 2013*). Aconitase activity assays were performed as previously described in *Graham et al. (2010)*; *Zhang et al. (2013)*.

## Dissection, immunostaining and lipid droplet staining by Nile Red

For muscle or NMJ immunostaining, dissection and immunostaining of third instar larvae were performed as described in *Bellen and Budnik (2000)*. Briefly, third instar larvae were fixed in 3.7% formaldehyde for 20 min at room temperature and washed in 0.4% Triton X-100. Primary antibodies were used at the following dilutions: mouse anti- ATP5A 1:500 (Abcam, Cambridge, MA), chicken anti-GFP 1:1000 (Abcam, Cambridge, MA), mouse anti-DLG 1:250 (DSHB, [*Parnas et al., 2001*]), guinea pig anti-EPS15 1:2000 (*Koh et al., 2007*), mouse anti-BRP 1:1000 (*Wagh et al., 2006*), rabbit anti-α-adaptin 1:500 (*Gonzalez-Gaitan and Jackle, 1997*), mouse anti-Glutamate receptor IIa (DSHB, Iowa City, IA, [*Schuster et al., 1991*]), guinea pig anti-Dap160 1:500 (*Roos and Kelly, 1998*), rabbit anti-HRP 1:1500 (Jackson ImmunoResearch, West Grove, PA), guinea pig anti-endophilin 1:200 (*Verstreken et al., 2002*), rabbit anti-synaptojanin (*Verstreken et al., 2003*), and rabbit anti-*Drosophila* vesicular glutamate transporter (DVGlut) 1:2000 (*Daniels et al., 2004*). Alexa 488 conjugated (Invitrogen), and Cy3 or Cy5 conjugated secondary antibodies (Jackson ImmunoResearch, West Grove, PA) were used at 1:250. Samples were mounted in VECTASHIELD (Vector Labs, Burlingame, CA).

For Lipid Droplet staining, third instar larvae were dissected in cold PBS and fixed in 4% paraformaldehyde for 30 min. Larvae were rinsed several times with 1× PBS to remove fixative and incubated for 10 min at 1:1000 dilution of PBS with 1 mg/ml Nile Red (Sigma, St. Louis, MO). Subsequently the tissues were rinsed with PBS and immediately covered with VECTASHIELD (Vector Labs, Burlingame, CA) for same-day imaging.

All confocal figures were acquired with confocal microscope (LSM510; Zeiss) using Plan Apochromat 40 × NA 1.4 and Plan Apochromat 63 × NA 1.4 objectives (Zeiss), followed by processing in LSM software (Zeiss), ImageJ, and Photoshop (Adobe).

## Electrophysiology and FM-143 labeling

Larval electrophysiological recordings were performed as described in *Koh et al. (2004)*. For labeling the exo-endo cycling pool (ECP) of vesicles, FM1-43 assays were performed as described (*Verstreken et al., 2005*, *2008*). Live images were acquired using a 40× water immersion lens and a Zeiss LSM510 confocal microscope.

## Ecdysteroid (20E) titers

Ecdysteroid levels were quantified by ELISA following the procedure described by *Porcheron et al. (1976)*, and adapted by *Pascual et al. (1995)*. For sample preparation, 20 to 30 staged larvae were weighed and preserved in 600 µl of methanol. Prior to the assay, samples were homogenized and centrifuged (10 min at 18,000×*g*) twice and the resultant methanol supernatants were combined and dried. Samples were resuspended in 50 µl of enzyme immunoassay (EIA) buffer (0.4 M NaCl, 1 mM EDTA, 0.1% BSA in 0.1 M phosphate buffer). 20E (Sigma, St. Louis, MO) and 20E-acetylcholinesterase (Cayman Chemical, Ann Arbor, MI) were used as the standard and enzymatic tracer. Absorbance was read at 450 nm using a FLUOstar Optima Spectrophotometer (BMG Labtech), results are expressed as 20E equivalents.

## Acknowledgements

We thank the Bloomington *Drosophila* Stock Center for flies and the Developmental Studies Hybridoma Bank for antibodies. We are grateful to Dr GW Dorn (*UAS-MFN1* and *UAS-MFN2*), Dr AJ Whitworth (Marf antibody) and Dr M Guo (*UAS-Opa IR* and *UAS-Drp1*) for fly stocks and antibodies. We thank Drs MF Wangler, ES Seto, KL Schulze and K Vankatachalam for critical reading. We thank H Pan and Y He for injections. We thank Dr S Jaiswal with help with qPCR of DRP1. Confocal microscopy at BCM is supported by the Intellectual and Developmental Disabilities Research Center (NIH 5P30HD024064). HS was supported by NIH 5R01GM067858, NIH T32 NS043124-11 and the Research Education and Career Horizon Institutional Research and Academic Career Development Award Fellowship 5K12GM084897. VB was supported by the NIH (5T32HD055200) and the Edward and Josephine Hudson Scholarship Fund. BX was supported by the Houston Laboratory and Population Science Training Program in Gene–Environment Interaction from the Burroughs Wellcome Fund (Grant No. 1008200). W-LC was supported by Taiwan Merit Scholarships Program sponsored by the National Science Council (NSC-095-SAF-I-564-015-TMS). SY was supported by a fellowship from the Nakajima Foundation and the Jan and Dan Duncan Neurological Research Institute at Texas Children's Hospital. This project was funded in part by NIH RC4GM096355-01 and gifts from the Robert A and Renee E Belfer Family Foundation, the Huffington Foundation, and Target ALS. HJB is an Investigator of the Howard Hughes Medical Institute.

## Additional information

### Funding

| Funder | Grant reference number | Author |
| --- | --- | --- |
| National Institutes of Health | RC4GM09355-01 | Hugo J Bellen |
| Belfer Center for Science and International Affairs, Harvard University | | Hugo J Bellen |
| Huffington Foundation | | Hugo J Bellen |
| Project A.L.S | Target ALS | Hugo J Bellen |
| Howard Hughes Medical Institute | | Hugo J Bellen |
| National Institutes of Health | 5R01GM067858 | Hector Sandoval |
| National Institutes of Health | NS043124-11 | Hector Sandoval |

| Funder | Grant reference number | Author |
|---|---|---|
| Research Education and Career Horizon Institutional Research and Academic Career Development Award Fellowship | 5K12GM084897 - Baylor College of Medicine | Hector Sandoval |
| National Institutes of Health | 5T32HD055200 | Vafa Bayat |
| Edward J. and Josephine G. Hudson Scholarship | | Vafa Bayat |
| Burroughs Wellcome Fund | 1008200 | Bo Xiong |
| National Science Council Taiwan | Taiwan Merit Scholarship Program, NSC-095-SAF-I-564-015-TMS | Wu-Lin Charng |
| Heiwa Nakajima Foundation | | Shinya Yamamoto |
| Texas Children's Hospital | Jan and Dan Duncan Neurological Research Institute | Shinya Yamamoto |

The funders had no role in study design, data collection and interpretation, or the decision to submit the work for publication.

## Author contributions

HS, Conception and design, Acquisition of data, Analysis and interpretation of data, Drafting or revising the article; C-KY, Made the Marf-transgenic and RNAi lines (Marf, Drp1 and dmiro) and established NMJ confocal microscopy assays; KC, Made the Marf-transgenic and established NMJ confocal microscopy assays; MJ, Designed and performed Drosophila X-chromosome screen (mutagenesis and mapping), Provided advice and insight during different stages of the project; TD, Performed the ETC enzymatic and aconitase assay; YQL, Performed the electrophysiology assays; VB, BX, KZ, GD, W-LC, SY, Designed and performed Drosophila X-chromosome screen (mutagenesis and mapping), Drafting or revising the article; LD, Performed sections and stains for TEM; BHG, Provided advice and insight during different stages of the project; HJB, Conception and design, Analysis and interpretation of data, Drafting or revising the article

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
