## [Decision Letter]

Thank you for sending your work entitled “Mitochondrial fusion but not fission regulates steroid hormone production, larval growth, and synaptic development” for consideration at *eLife*. Your article has been favorably evaluated by K VijayRaghavan (Senior editor), Richard Youle (Reviewing editor), and 3 reviewers, one of whom, Bingwei Lu, has agreed to reveal his identity.

The Reviewing editor and the reviewers discussed their comments before we reached this decision, and the Reviewing editor has assembled the following comments to help you prepare a revised submission.

1) Is the deficit of ecdysone production in RG indeed responsible for the NMJ morphology defects in dMarf mutants? Does ecdysone supplementation rescue the extended larval lifespan and NMJ defects?

2) The direct comparison of Marf and Drp1 phenotypes is really only valid if the authors are comparing severe loss-of-function alleles of these genes. While their work suggests that this is the case for Marf, it may not be true of the Drp1 allele they have chosen for study. The Drp1[2] allele produces numerous adult escaper homozygotes. By contrast, other Drp1 alleles (e.g., T26, KG) do not. The authors also provide no data to demonstrate how effectively their RNAi constructs against Drp1 and Opa1 knock down expression of the intended targets. The concern is that some of the phenotypic differences reported may be more reflective of the fact that the authors are comparing a severe loss-of-function allele of MARF to a weak hypomorphic allele of DRP1. The authors need to address this matter.

3) Do the mutant effects of Marf and Opa1 on LD formation and ecdysone synthesis involve their canonical function in mitochondrial fission/fusion dynamics or some non-canonical function? Although Drp1 loss of function (LOF) was used to show lack of effects by fission, but since Drp1 LOF and Marf or Opa1 LOF exert opposite effects on mitochondrial morphology, a key control is Drp1 overexpression. If Drp1 overexpression in RG does not have the same effect as dMarf LOF, then one could conclude that some non-canonical function of Marf might be involved. Also, do LOF Drp1 alleles compensate for the LOF dMarf phenotype?

4) The authors show that hMFN1 and hMFN2 are both required for full rescue of the Marf phenotypes, but they do not clearly delineate the roles played by these two proteins. One possibility is that hMFN2 regulates mitochondrial organelle contacts, whereas hMFN1 regulates mitochondrial morphology. However, the authors do not address these possibilities directly. This matter could be investigated by using TEM to examine mitochondrial organelle contacts in the ring glands of Marf mutants expressing Mfn1 or Mfn2, and by monitoring mitochondrial morphology in larval body wall muscles from these same genotypes. These experiments seem warranted in light of the partially overlapping work of Debattisti, et al.

5) The authors focus on MarfB, but their molecular evidence makes a stronger case that MarfG is a null allele, so they need to provide rationale for this decision. Also, they use MarfB in all of their experiments except for those in Figure 4; why the switch in alleles for the physiology? Finally, the authors sometimes show data with WT controls (e.g., Figure 1, Figure 2, etc.) and others using genomic rescued flies as the “WT control” (e.g., Figure 3, Figure 4, etc.); indeed, the authors state in text that “response to low frequency stimulations in Marf mutants are not different than WT controls” but their “WT controls” are really Marf mutants rescued with a transgene, which is not really a WT control.

---

## [Author Response]

*1) Is the deficit of ecdysone production in RG indeed responsible for the NMJ morphology defects in dMarf mutants? Does ecdysone supplementation rescue the extended larval lifespan and NMJ defects*?

We were unable to rescue both extended larval lifespan and NMJ defects in *Marf*^*B*^ mutants by feeding them 20-hydroxyecdysone (20E) at different concentrations (0.0625, 0.125, 0.250 and 0.50mM) started on different days of the *Marf*^*B*^ mutants extended larval lifespan. In all conditions the addition of 20E resulted in the larvae climbing out of the food and dying within 12 to 18 hours. However, we were able to rescue both the lifespan and NMJ phenotypes in third instar larvae with a RG specific knock down of *Marf.* These results have now been included in Figure 6—figure supplement 1.

*2) The direct comparison of Marf and Drp1 phenotypes is really only valid if the authors are comparing severe loss-of-function alleles of these genes. While their work suggests that this is the case for Marf, it may not be true of the Drp1 allele they have chosen for study. The Drp1[2] allele produces numerous adult escaper homozygotes. By contrast, other Drp1 alleles (e.g., T26, KG) do not. The authors also provide no data to demonstrate how effectively their RNAi constructs against Drp1 and Opa1 knock down expression of the intended targets. The concern is that some of the phenotypic differences reported may be more reflective of the fact that the authors are comparing a severe loss-of-function allele of MARF to a weak hypomorphic allele of DRP1. The authors need to address this matter*.

We thank the reviewers for raising this question and attempted to use homozygous combination of both *Drp1*^*KG38015*^ and *Drp1*^*[T26]*^ alleles but unfortunately the alleles die as first instars. Transheterozygous combinations of *Drp1*^*KG38015*^, *Drp1*^*[T26]*^ or *Drp1*^*1*^ with *Drp1*^*2*^ resulted in third instar larval or pupal lethality and no adult escapers. These stronger allelic combinations allow us to compare more severe loss-function genotypes of *Drp1* to *Marf* mutants. All combinations of *Drp1* transheterozygous alleles have similar phenotypes as *Drp1*^*2*^ homozygous mutants with respect to mitochondrial morphology, mitochondrial membrane potential, ATP levels, ROS intensity, bouton numbers and 20-hydroxyecdysone (20E) levels. We have now included all these results in Figure 2–figure supplement 1.

We agree with the reviewers that the efficacy of Drp1 and Opa1 RNAi lines needs to be addressed. We have now included in the text and figure legend of Figure 5 that the efficiency of the Drp1 RNAi (82% knockdown using a ubiquitous driver Actin-Gal4) and have also added a reference showing the efficiency of the Opa RNAi line that we use in our study (56).

*3) Do the mutant effects of Marf and Opa1 on LD formation and ecdysone synthesis involve their canonical function in mitochondrial fission/fusion dynamics or some non-canonical function? Although Drp1 loss of function (LOF) was used to show lack of effects by fission, but since Drp1 LOF and Marf or Opa1 LOF exert opposite effects on mitochondrial morphology, a key control is Drp1 overexpression. If Drp1 overexpression in RG does not have the same effect as dMarf LOF*, *then one could conclude that some non-canonical function of Marf might be involved. Also, do LOF Drp1 alleles compensate for the LOF dMarf phenotype?*

We now show that overexpression of Drp1 in RG does not alter NMJ morphology, LD numbers and 20-hydroxyecdysone (20E) levels suggesting a non-canonical role of Marf in the RG. We have included these data in Figure 6—figure supplement 1.

LOF of *Drp1* does not compensate for LOF Marf phenotypes in mitochondrial morphology, mitochondrial membrane potential, ATP levels, ROS intensity, bouton numbers and 20E levels. These data provide compelling evidence that LOF *Drp1* alleles do not compensate for LOF Marf phenotypes. We have included these data in Figure 2–figure supplement 1B.

*4) The authors show that hMFN1 and hMFN2 are both required for full rescue of the Marf phenotypes, but they do not clearly delineate the roles played by these two proteins. One possibility is that hMFN2 regulates mitochondrial organelle contacts, whereas hMFN1 regulates mitochondrial morphology. However, the authors do not address these possibilities directly. This matter could be investigated by using TEM to examine mitochondrial organelle contacts in the ring glands of Marf mutants expressing Mfn1 or Mfn2, and by monitoring mitochondrial morphology in larval body wall muscles from these same genotypes. These experiments seem warranted in light of the partially overlapping work of Debattisti, et al*.

We thank the reviewers for raising this point and have added ultrastructural data of *Marf*^*B*^ RG with expression of human MFN1 or MFN2. The data illustrates that MFN2 restores the loss of ER/Mito contacts in *Marf*^*B*^ mutants similar to controls. On the other hand, expression of MFN1 in *Marf* mutants does not restore ER/Mito contacts. These data are now provided in Figure 7—figure supplement 1.

We also performed a comparison of mitochondrial morphology in *Marf*^*B*^ mutant muscles with expression of human MFN1 or MFN2. Neither expression of MFN1 or MFN2 fully restored mitochondrial morphology when compared to controls, suggesting that both MFN1 and MFN2 are both required to restore mitochondrial morphology. This is now added to Figure 7—figure supplement 2.

*5) The authors focus on MarfB, but their molecular evidence makes a stronger case that MarfG is a null allele, so they need to provide rationale for this decision. Also, they use MarfB in all of their experiments except for those in*
Figure 4*; why the switch in alleles for the physiology? Finally, the authors sometimes show data with WT controls (e.g.,*
Figure 1*,*
Figure 2*, etc.) and others using genomic rescued flies as the “WT control” (e.g.,*
Figure 3*,*
Figure 4*, etc.); indeed, the authors state in text that “response to low frequency stimulations in Marf mutants are not different than WT controls” but their “WT controls” are really Marf mutants rescued with a transgene, which is not really a WT control*.

Based on Western blots, the expression in *Marf*^*G*^ is higher (51%) than *Marf*^*B*^ (4%). This suggests that the stop codon in *Marf*^*G*^ can be a read through stop codon and produce some protein. Based on this data, we decided to use *Marf*^*B*^, which is also in a similar range to the RNAi knockdown of *Marf* (11%). The data are now presented in Figure 1—figure supplement 1.

For the electrophysiological experiments, *Marf*^*B*^ mutants are too small in size and we cannot obtain size matched larvae that have a resting membrane potential lower than -60mV. Therefore, we decided to use hemizygous *Marf*^*E*^ and transheterozygous *Marf*^*B*^*/Marf*^*E*^ allelic combinations in Figure 4. We have now clarified this point in the text. Finally, we also made the suggested corrections and use “Marf Genomic-rescue controls”, rather than WT controls.